# Discovery of Indole–Thiourea Derivatives as Tyrosinase Inhibitors: Synthesis, Biological Evaluation, Kinetic Studies, and In Silico Analysis

**DOI:** 10.3390/ijms25179636

**Published:** 2024-09-05

**Authors:** Yang Xu, Xuhui Liang, Chang-Gu Hyun

**Affiliations:** 1Jeju Inside Agency and Cosmetic Science Center, Department of Chemistry and Cosmetics, Jeju National University, Jeju 63243, Republic of Korea; iamxuyang1990@gmail.com (Y.X.); lxh03036@naver.com (X.L.); 2Department of Beauty and Cosmetology, Jeju National University, Jeju 63243, Republic of Korea

**Keywords:** tyrosinase, hyperpigmentation, indole–thiourea derivatives, pharmacokinetics, molecular docking, molecular dynamics simulations

## Abstract

Tyrosinase, a key enzyme in melanin synthesis, represents a crucial therapeutic target for hyperpigmentation disorders due to excessive melanin production. This study aimed to design and evaluate a series of indole–thiourea derivatives by conjugating thiosemicarbazones with strong tyrosinase inhibitory activity to indole. Among these derivatives, compound **4b** demonstrated tyrosinase inhibitory activity with an IC_50_ of 5.9 ± 2.47 μM, outperforming kojic acid (IC_50_ = 16.4 ± 3.53 μM). Kinetic studies using Lineweaver–Burk plots confirmed competitive inhibition by compound **4b**. Its favorable ADMET and drug-likeness properties make compound **4b** a promising therapeutic candidate with a reduced risk of toxicity. Molecular docking revealed that the compounds bind strongly to mushroom tyrosinase (mTYR) and human tyrosinase-related protein 1 (TYRP1), with compound **4b** showing superior binding energies of −7.0 kcal/mol (mTYR) and −6.5 kcal/mol (TYRP1), surpassing both kojic acid and tropolone. Molecular dynamics simulations demonstrated the stability of the mTYR−**4b** complex with low RMSD and RMSF and consistent Rg and SASA values. Persistent strong hydrogen bonds with mTYR, along with favorable Gibbs free energy and MM/PBSA calculations (−19.37 kcal/mol), further support stable protein–ligand interactions. Overall, compound **4b** demonstrated strong tyrosinase inhibition and favorable pharmacokinetics, highlighting its potential for treating pigmentary disorders.

## 1. Introduction

In skin-brightening research, tyrosinase inhibition is essential due to its pivotal role in melanin biosynthesis [1]. Tyrosinase catalyzes the hydroxylation of monophenols (e.g., L-tyrosine) into *o*-diphenols (e.g., L-DOPA), which are then oxidized to *o*-quinones (e.g., *o*-dopaquinone). These reactive *o*-quinones either polymerize to form dark pigments or conjugate with other molecules, enhancing pigment intensity [2]. Increased pigmentation can contribute to various skin disorders, such as hyperpigmentation, melasma, and age spots, potentially leading to aesthetic concerns and psychological effects [3,4].

Tyrosinase inhibitors are categorized by their binding mechanisms—competitive (binding at the active site, competing with substrates), noncompetitive (binding elsewhere, altering enzyme conformation or activity), and uncompetitive (affecting enzyme activity or synthesis) [5,6]. So far, an extensive range of tyrosinase inhibitors have been identified, including polyphenolic compounds (flavonoids, coumarins) [7], benzaldehyde and benzoate derivatives [8], long-chain lipids and steroids [9], various natural and synthetic inhibitors, and irreversible inactivators [6]. These inhibitors critically modulate tyrosinase activity and substrate reactions, proving valuable not only in skin brightening but also in exploring new avenues for treating various skin disorders [10].

Indole derivatives play a crucial role in melanin formation, including dopamine (DOPA) [11] and its oxidized products, such as 5,6-dihydroxyindole (DHI) [12], indolequinone (IQ), and dihydroxyindolecarboxylic acid (DHICA), as well as 5,6-indolequinone-2-carboxylic acid (IQCA) [13]. These compounds interact with tyrosinase, influencing its catalytic activity and thereby regulating melanin production. Indole derivatives not only inhibit multiple enzymes (including tyrosinase, tyrosinase-related protein 1, and dopachrome tautomerase) but also exhibit diverse biological activities, such as anti-inflammatory [14,15], antioxidant [16], and anticancer properties [17,18,19]. The structural resemblance between natural indole intermediates and synthetic indole derivatives provides a solid foundation for designing effective tyrosinase inhibitors. These inhibitors can potentially disrupt substrate recognition or catalytic activity at the enzyme’s active site, thereby impeding melanin formation [20,21].

The synthesis of derivatives based on an indole core presents a promising strategy for developing tyrosinase inhibitors (Figure 1). Natural products such as indole-3-carbaldehyde [22] and indole-3-ethanol (tryptophol) [23] have exhibited significant tyrosinase inhibition activity, while indole-3-carboxylic acid [24] and indole-3-acetic acid [25] were shown to have inhibitory effects on B16F10 melanoma cells in previous studies. Reported syntheses of indole derivatives with tyrosinase inhibitory activity typically involve derivatization at positions 1, 2, 3, and 5 of the indole core. Substitution at position 1 often involves benzyl groups [20], while position 2 is linked to phenol via an amide bond or other active groups [26]. Position 3 serves as a crucial branch point for introducing nitrogen-containing heterocycles known for their pharmacological effects, whereas electron-donating or -withdrawing substituents are commonly introduced at position 5 [27,28,29]. Additionally, the thiosemicarbazone group represents a critical moiety for tyrosinase inhibition. Structure–activity relationship (SAR) analyses indicate that the carbon–sulfur double bond is pivotal for activity [30]. An unsubstituted amino group at the outermost end of the thiourea moiety enhances inhibition, while further introduction of amino groups diminishes inhibition [31,32]. The introduction of methyl substituents on unsaturated carbons in imine structures enhances activity compared with that of unsubstituted forms [33], while prolonged alkyl chains negatively impact inhibition under in vitro conditions; however, in B16 mouse melanoma cells, the trends of the inhibition of melanin synthesis are contradictory [34].

Melanin biosynthesis requires three essential enzymes: tyrosinase (TYR), tyrosinase-related protein 1 (TYRP1), and tyrosinase-related protein 2 (TYRP2) [35]. Mushroom tyrosinase (mTYR) is commonly used to screen tyrosinase inhibitors due to its availability and ease of use, despite its limited sequence homology with human tyrosinase (only 12% sequence identity). In contrast, human tyrosinase (hTYR) shares a higher sequence homology with human-derived proteins (40% sequence identity) [36,37,38]. Due to the complexity, flexibility, and high glycosylation of human tyrosinase, crystallization is extremely challenging, leading to contradictory findings regarding its physiological functions [39]. Homology modeling of tyrosinase-related protein 1 (PDB ID: 5M8M) revealed significant structural differences from mushroom tyrosinase (PDB ID: 2Y9W), with an RMSD of 11.477 and binding site shifts [40]. Structural differences between human and mushroom tyrosinase imply distinct molecular characteristics for effective inhibitors [41,42]. Previous molecular docking studies on mushroom tyrosinase identified residues N81, M280, and N260 as important for substrate binding, with H263 forming a π interaction [43,44,45]. In contrast, studies on tyrosinase-related protein 1 have revealed key residues involved in ligand binding. The crystal structures of TYRP1 with various bound ligands have shown that Y362, R374, H381, S394, and T391 are crucial for ligand binding [46]. These findings provide a valuable foundation for drug design targeting human tyrosinase.

In this study, the tyrosinase inhibitor indole-3-carbaldehyde was utilized as a starting material to synthesize a series of indole–thiourea derivatives containing thiosemicarbazone moieties at the 3-position. These compounds were screened for tyrosinase inhibition activity, followed by kinetic analysis to elucidate the inhibition mechanism. Molecular docking studies were employed to further investigate the interaction between active compounds and tyrosinase, while molecular dynamics (MD) simulations revealed the stability and strength of these interactions, which accounted for protein flexibility.

## 2. Results and Discussion

### 2.1. Chemistry

Indole–thiourea derivatives **4a**–**4k** were synthesized using a convergent approach, as outlined in Figure 1. The structural elucidation of all synthesized compounds was confirmed through comprehensive spectroscopic analyses, including ^1^H NMR, HRMS, and HPLC. Initially, intermediates **2f** and **2g** were prepared via nucleophilic substitution of halogenated alkanes with indole-3-carbaldehyde (**1a**). The ^1^H NMR spectra of these intermediates revealed specific chemical shifts that were indicative of their structures, intermediate **2f** displayed a quartet at *δ* 4.30 (q, *J* = 7.3 Hz, 2H) and a triplet at *δ* 1.42 (t, *J* = 7.2 Hz, 3H), which corresponded to the ethyl group, while intermediate **2g** showed a singlet at *δ* 3.89 (s, 3H) for the methyl group. Subsequently, intermediates **2a–k** (indole-3-carbaldehyde derivatives) underwent nucleophilic addition with thiosemicarbazone, forming Schiff bases and resulting in the target indole–thiourea derivatives **4a–4k** at reasonable yields (80–89%). As a representative example, compound **4b** was isolated as a yellow solid with an 84% yield and a melting point of 207–209 °C. The ^1^H NMR spectrum of **4b** revealed characteristic signals: a singlet at 11.70 ppm for the indole NH proton, doublets at 7.42 ppm (*J* = 8.8, 4.6 Hz) and 7.98 ppm (*J* = 5.1, 2.3 Hz) for the thiourea NH_2_ protons, a singlet at 11.13 ppm for the thiourea NH proton, and signals for the indole core protons at 7.03 ppm (td, *J* = 9.2, 2.6 Hz, H6), 7.58 ppm (s, H4), 7.87 ppm (s, H7), 8.01 ppm (d, *J* = 2.6 Hz, H2), and 8.27 ppm (s, H10). The ^13^C NMR spectrum (101 MHz, DMSO-*d6*) of the compound revealed a thiocarbonyl group at *δ* 176.54 ppm (C11). The doublet peaks at *δ* 158.0 ppm (*J* = 234.3 Hz, C5) confirmed a direct C-F bond. Doublet peaks at *δ* 112.80 ppm (*J* = 10.0 Hz, C6) and *δ* 111.32 ppm (*J* = 4.0 Hz, C4) indicated ortho coupling, with *J*_*G**E**M*_ greater than *J*_*o*_. In indole–thiourea derivatives, doublets are also observed at *δ* 124.15 ppm (*J* = 11.1 Hz, C9), *δ* 110.83 ppm (*J* = 26.0 Hz, C3), and *δ* 107.12 ppm (*J* = 24.0 Hz, C7), thereby highlighting both direct and indirect C-F couplings [47]. The carbon at *δ* 140.69 ppm (C10) corresponds to an imine carbon, while the carbons at *δ* 133.70 ppm (C8) and *δ* 132.80 ppm (C2) are attributed to the indole ring. HRMS analysis further confirmed the structure of **4b** with a calculated molecular ion (M+H)^+^ of 237.0610, which matched the observed ion at *m/z* 237.0611 (M+H)^+^. Under identical HPLC detection conditions and using the same analytical method, all synthesized compounds exhibited integration area percentages greater than 90%. Compound **4b** showed a retention time of 22.529 min and an area percentage of 97.21%. These spectroscopic results validated the successful synthesis and structural assignment of compound **4b**, supporting the efficacy of our synthetic strategy and characterization methods.

The stability of indole–thiourea derivatives with imine structures in aqueous solution is crucial for drug design, as their hydrolysis and reactivity influence efficacy, safety, and pharmacokinetics [48]. The synthesized compounds, dissolved in aqueous solution and stored at room temperature, showed better stability after 2 and 4 days. After two months, degradation was observed in all compounds, but only compound **4i** exhibited significant degradation (main peak area at 65%), while others remained above 90% (Appendix A). Most compounds containing thiosemicarbazone groups in aqueous solution showed some stability, likely due to the increased stability of conjugated imines formed between the imine and indole double bonds, as conjugated imine structures are commonly observed in marine alkaloids [49]. This enhanced stability may also be due to the π–electron delocalization effect or the inductive effect of X_2_ = N or O [50], which contribute to the greater stability of hydrazones compared to imines.

### 2.2. Biological Evaluations

#### 2.2.1. Anti-Tyrosinase Activity and Structure–Activity Relationship (SAR)

The synthesized compounds (**4a**–**4k**) were evaluated for their tyrosinase inhibitory activity using IC_50_ values, with kojic acid as a positive control. Table 1 summarizes the results. All compounds displayed varying degrees of inhibitory activity, with IC_50_ values ranging from 5.9 to 163.3 µM. In comparison, kojic acid had an IC_50_ value of 16.4 ± 3.53 µM. The overall structure of a molecule and its specific substituents influence inhibitory activity, revealing structure–activity relationships (SARs). Variations in the indole group substituents significantly affected the inhibitory potential of these compounds.

We summarize the findings on the structure–activity relationships (SARs) in the following. First, the effect of halogen substitution: The inhibitory activity was influenced by the type of halogen substitution (e.g., -F, -Cl, -Br), as observed in compounds **4b**–**4d**. Specifically, compound **4b** (IC_50_ = 5.9 ± 2.47 μM) exhibited the highest inhibitory activity, followed by **4c** (IC_50_ = 13.2 ± 1.68 μM) and **4d** (IC_50_ = 14.9 ± 2.86 μM). The significant activity of **4b** suggested that smaller halogen atoms, such as fluorine, enhanced the enzyme interactions and inhibitory potency. Compounds with electron-withdrawing groups on the benzene ring demonstrated superior inhibitory activity to those with electron-donating groups. Specifically, compounds with electron-withdrawing groups at position 5 (**4b**, **4c**, **4d**) showed better activity than those substituted with electron-donating groups, such as **4e** (R_2_ = OCH_3_), **4j** (R_2_ = COOCH_3_), and **4k** (R_2_ = OH). However, with a methyl group at the indole 1-position, compound **4i** (R_1_ = CH_3_, R_2_ = Br) showed superior activity to that of **4h** (R_1_ = CH_3_, R_2_ = OCH_3_).

Second, the effect of the indole amino group: Compounds **4g** (R_2_ = H), **4h** (R_2_ = OCH_3_), and **4i** (R_2_ = Br) with a methyl group at the amino position exhibited significantly lower inhibitory activity than non-substituted compounds **4a** (R_2_ = H), **4e** (R_2_ = OCH_3_), and **4d** (R_2_ = Br). The unsubstituted amino group acts as a hydrogen bond donor, facilitating tight binding to the protein, which enhances the inhibitory activity [51].

Third, the effect of alkyl chain length on the indole amino group: We further investigated the effect of the length of the alkyl chain connected to the indole amino group. As the carbon chain’s length increased, the molecular hydrophobicity and inhibitory activity also increased. Compound **4f** (R_1_ = CH_2_CH_3_) outperformed **4g** (R_1_ = CH_3_). Previous MD simulations and free energy calculations indicated that longer carbon chains enhance protein–ligand binding free energy by increasing hydrophobic contact and reducing the desolvation energy [52]. Additionally, longer carbon chains improve spatial filling, induced fit, and molecular flexibility [53,54]. These findings underscore the importance of specific substituents and structural modifications in optimizing the inhibitory activity of the compounds under study.

#### 2.2.2. Determination of the Inhibitory Mechanism through Enzyme Kinetics

Due to the potent inhibition of mushroom tyrosinase by compound **4b**, kinetic studies were conducted using Lineweaver–Burk double reciprocal plots to elucidate its inhibitory mechanisms. Three concentrations (0 μM, 3 μM, 6 μM, and 12 μM) of the indole derivative **4b** in the presence of varying L-tyrosine concentrations (0.1, 0.2, 0.4, 0.6, 0.8, 1 mM) were employed for the kinetic analyses. The results are summarized in Figure 2. The Lineweaver–Burk plots generated four distinct lines with different slopes corresponding to each concentration of the derivative. All lines converged at a single point on the y-axis, indicating that the Vmax value of compound **4b** (0.0135 mM/min) was consistent across the concentrations. Additionally, the data revealed that the K_M_ values of **4b** increased in a concentration-dependent manner, implying the competitive inhibition of mTYR alongside L-tyrosine binding. Specifically, the K_M_ values were 1.515, 7.846, 15.740, and 23.586 mM at 0, 3, 6, and 12 μM of **4b**, respectively. Furthermore, the kinetic studies yielded K_i_ values of 1.68 × 10^−7^, 5.68 × 10^−8^, and 2.06 × 10^−8^ M at 3, 6, and 12 μM of **4b**, indicating the potency of inhibition at these concentrations.

#### 2.2.3. Free Radical-Scavenging Activity

Radical-scavenging assays evaluate a compound’s capacity to neutralize free radicals, providing insight into its potential to reduce oxidative damage caused by tyrosinase activity [55]. The DPPH radical-scavenging activity of the eleven synthesized compounds (**4a**–**4k**) and a reference compound, ascorbic acid, was evaluated. The SC_50_ values, which represent the concentration required to scavenge 50% of the DPPH radicals, were determined and are summarized in Table 2 and Appendix A. The SC_50_ values indicated that compound **4k** exhibited the most potent DPPH radical-scavenging activity with an SC_50_ of 24.6 ± 3.7 μM, followed closely by ascorbic acid with an SC_50_ of 37.9 ± 0.3 μM. This suggested that compound **4k** had a comparable or slightly superior antioxidant capacity relative to that of ascorbic acid. Compounds **4j** and **4h** exhibited moderate radical scavenging activity, with SC_50_ values of 275.7 ± 2.7 μM and 340.2 ± 3.1 μM, respectively. On the other hand, compounds **4a**, **4b**, **4c**, **4e**, and **4g** showed relatively weaker activity, with SC_50_ values ranging from 482.7 ± 1.3 μM to 659.2 ± 0.5 μM. Compounds **4d** and **4f**, with SC_50_ values of 769.0 ± 2.6 μM and >1000 μM, respectively, exhibited the least effective antioxidant activity among the tested compounds. 

The data indicate that structural differences significantly influenced the antioxidant activities of the synthesized compounds. Hydroxyl groups in phenolic compounds act as antioxidants by donating hydrogen atoms, thereby neutralizing free radicals and stabilizing them through resonance and electron delocalization [56,57]. This prevents oxidative damage and maintains cellular stability. Compound **4k**’s high activity was likely due to the hydroxyl group at the 5-position of the indole–benzene ring. Ascorbic acid, with its unsaturated enediol structure, also showed strong antioxidant capacity. Both compound **4k** and ascorbic acid share crucial hydroxyl groups, suggesting a common feature for enhancing radical scavenging efficiency.

### 2.3. In Silico Experiments

#### 2.3.1. Molecular Properties and Drug Likeness

The results of ADMET (absorption, distribution, metabolism, excretion, and toxicity) analysis showed that compound **4b** exhibited several pharmacokinetic and toxicological advantages over kojic acid and tropolone (Appendix A). It had a higher LD_50_ (3.1502 mol/kg), indicating lower acute toxicity than kojic acid (LD_50_ = 2.0673 mol/kg) and tropolone (LD_50_ = 2.4899 mol/kg). With a 97.0% plasma protein-binding rate, compound **4b** ensures prolonged drug action and reduced side effects. It selectively inhibits only CYP1A2, minimizing drug–drug interactions. Importantly, compound **4b** does not cross the blood–brain barrier, thereby reducing side effects in the central nervous system (CNS) [58]. It also shows no hepatotoxicity, Ames toxicity, skin sensitization, or hERG inhibition. Despite its moderate permeability, compound **4b** achieves high human intestinal absorption (89.027%), ensuring effective bioavailability. In conclusion, compound **4b** offers a more favorable safety profile with higher acute toxicity tolerance, extensive plasma protein binding, minimal CYP450 inhibition, and no CNS penetration in comparison with kojic acid and tropolone. These characteristics make compound **4b** a promising candidate with potentially lower risks of systemic toxicity and drug–drug interactions, resulting in a safer pharmacokinetic profile for therapeutic applications.

In drug-likeness analysis (Appendix A), all compounds showed superior drug likeness to kojic acid and tropolone, adhering to all key drug-likeness rules: RO5 [59], the Ghose Filter [60], the Veber Rule [61], and the Egan Rule [62]. This comprehensive compliance indicates that compound **4b** possesses excellent pharmacokinetic properties and a high potential for successful development as an orally bioavailable drug. Conversely, while kojic acid and tropolone are promising, their violations of the Ghose Filter criteria (three violations: MW < 160, MR < 40, atoms < 20) may pose challenges in drug development.

#### 2.3.2. Molecular Docking Simulation

The docked complexes of the synthesized ligands (**4a**–**4k**) with mushroom tyrosinase and human tyrosinase-related protein 1 were analyzed based on the minimum energy values and ligand interaction patterns. Tropolone and kojic acid served as co-crystallized inhibitors for polyphenol oxidase (PPO) family PPO3 protein (PDB ID: 2Y9X) and human tyrosinase-related protein 1 (PDB ID: 5M8M), respectively. The re-docking results of these co-crystallized ligands demonstrated that tropolone had a binding energy of −6.3 kcal/mol with mTYR, with an RMSD value of 1.565 Å between the pre- and post-docking conformations (Figure 3a). Kojic acid exhibited a binding energy of −5.9 kcal/mol with TYRP1, with an RMSD value of 1.935 Å (Figure 3b). 

Both co-crystallized ligands bound to the same site on their respective proteins with minimal conformational differences, achieving RMSD values that were lower than the widely accepted validation standard of 2 Å [63,64], thereby validating the docking methodology.

In this study, we conducted molecular docking analyses to investigate the binding affinities of 13 compounds (**4a**–**4k**, tropolone, and kojic acid) with mTYR and TYRP1. The results indicated that all synthesized ligands displayed favorable binding energy values and effectively interacted within the active site of the target proteins, as detailed in Table 3. A comparative analysis of the docking scores for both mTYR and TYRP1 revealed that compounds **4a** and **4b** exhibited strong binding to mTYR (−7.0 kcal/mol) but only moderate binding to TYRP1 (−5.8 and −6.5 kcal/mol). Conversely, compounds **4b**, **4d**, and **4f** showed similar high binding affinities for both mTYR and TYRP1. Compounds **4h** and **4k** exhibited nearly identical binding energies for mTYR and TYRP1, suggesting they may have similar binding modes and affinities for both proteins. Compound **4i** displayed a notable difference in binding energy between mTYR (−6.0 kcal/mol) and TYRP1 (−4.5 kcal/mol), indicating significant variability in the binding interactions.

In conclusion, the molecular docking results highlight the importance of specific structural features in determining the binding affinities of tyrosinase inhibitors. Compounds with higher binding energies, such as **4a**, **4b**, **4f**, and **4k**, could be promising candidates for further development as tyrosinase inhibitors.

Compound **4b** demonstrated the highest tyrosinase inhibitory activity and strong binding affinity in docking simulations with both mTYR and TYRP1, as shown in Figure 4a,b. In mTYR, the pyrrole amino group of compound **4b** forms a hydrogen bond with MET-280 (2.6 Å, green dashed line), while the indole structure interacts with VAL-283, HIS-263, and ALA-286 through Pi–sigma (purple dashed line), Pi–sulfur (pink dashed line), and Pi–Pi alkyl (light pink dashed line) interactions, respectively. The fluorine atom, due to its high electronegativity, attracts partial positive charges from amino acid side chains, forming weak hydrogen-like interactions with HIS-85 and HIS-259 (blue dashed line), which help stabilize the ligand–receptor binding [65,66,67]. The docking energy of compound **4b** with mTYR was −7.0 kcal/mol, which was better than that of the co-crystallized ligand tropolone (−6.3 kcal/mol) and the positive control kojic acid (−5.3 kcal/mol).

In TYRP1, compound **4b** forms a hydrogen bond with THR-391 (3.0 Å, green dashed line) and a carbon–hydrogen bond with TYR-362 via its imine group (Schiff base). The amino groups on both sides of the thiourea moiety form two hydrogen bonds with GLU-216 (2.4 Å, 2.5 Å), while the outer amino group of the thiourea forms a hydrogen bond with ASP-212 (2.4 Å). The fluorine on the indole benzene ring forms a hydrogen bond with ARG-374 (3.0 Å), indicating strong hydrogen bond stability. The pyrrole ring of the indole interacts with HIS-381 via Pi-Pi T-shaped and carbon–hydrogen bonds, and the indole benzene ring interacts with LEU-382 via Pi–alkyl interactions. The docking energy of compound **4b** with TYRP1 was −6.5 kcal/mol, which was superior to that of the co-crystallized ligand kojic acid (−5.9 kcal/mol). 

The detailed interactions of other compounds with mTYR and TYRP1 are provided in Appendix A. The results show that compound **4f**’s ethyl groups interact with a greater number of amino acid residues than methyl groups do, forming more interactions (e.g., alkyl, Pi–alkyl, Pi–sigma). These interactions allow ethyl groups to occupy more active sites than methyl groups do, thereby enhancing their inhibitory activity [68,69,70].

The reproducibility of the docking results was confirmed with ten repeated docking simulations for compound **4b** with both mTYR and TYRP1, showing binding energies within ±0.1 kcal/mol (Appendix A). These findings indicate that compound **4b** exhibits high binding affinity to both mTYR and TYRP1, suggesting a potential pharmacological effect that is similar to that of the co-crystallized ligands, likely impacting the structure, function, and bioactivity of tyrosinase proteins.

#### 2.3.3. Molecular Dynamics (MD) Simulation 

As previously mentioned [23], we conducted 100 ns MD simulations on the complexes of mTYR with the co-crystallized ligand (tropolone) and compounds to further assess binding stability and affinity. This was necessary because the semi-flexible docking method does not fully account for protein flexibility. This study analyzed the MD simulation trajectories of the complexes, focusing on the root mean square deviation (RMSD), root mean square fluctuation (RMSF), radius of gyration (Rg), number of hydrogen bonds (H-bonds) between the protein and compound, solvent-accessible surface area (SASA), Gibbs free energy landscape (FEL-energy), energy contribution of binding residues (Residue-energy), average binding free energy (Total-energy), and conformational structures of the complexes at 0 ns, 25 ns, 50 ns, 75 ns, and 100 ns of the MD simulation.

An RMSD curve was used to assess the protein–ligand complex stability; lower values indicate minimal structural changes and greater stability. The PPO3 protein complexed with compound **4b** showed consistently low RMSD fluctuations, closely mirroring the co-crystallized ligand’s curve, suggesting that the PPO3–**4b** complex is stable (Figure 5a).

The RMSF curve illustrates residue fluctuation in proteins during dynamic simulations, where higher values denote greater flexibility and lower values indicate stability. In the PPO3–**4b** complex (Figure 5b), the RMSF remains within 1 nm, with slight fluctuations (~0.4 nm) observed at residues 70–80 near the protein periphery. The RMSF curve of compound **4b** closely overlaps with that of the co-crystallized ligand group, suggesting a minimal impact on residue stability in PPO3 and affirming the complex’s overall stability.

The Rg assesses structural compactness and stability, with larger values indicating greater expansion in dynamic simulations and smaller values indicating stability. In the PPO3–**4b** complex (Figure 6a), the Rg curve stabilizes around 2.1 nm, demonstrating overall stability with minimal fluctuations. Compared to the co-crystallized ligand, PPO3–**4b** exhibits smaller Rg fluctuations, indicating the formation of a tight and stable complex with compound **4b**. Importantly, the inclusion of compound **4b** did not significantly alter the protein’s overall structure.

To investigate hydrogen bonding at the binding site of the complex, we computed the number of key hydrogen bonds stabilizing ligand–protein interactions. During 100 ns of dynamic simulation, stable hydrogen bonding interactions (1–3 bonds) formed between the PPO3 protein and compound **4b** (Figure 6b), indicating robust binding stability. In contrast, the co-crystallized ligand group maintained a stable hydrogen bond count of one throughout, suggesting slightly superior stability of the PPO3–**4b** complex compared with that of the co-crystallized ligand group.

The SASA is a factor in studying protein structures’ folding and stability. Proteins with stable structures typically exhibit consistent SASA curves. The SASA curve of the PPO3–**4b** complex shown in Figure 7 shows stable fluctuations throughout, without significant changes. Moreover, the SASA curve of the co-crystallized ligand group closely overlaps with that of the PPO3–**4b** group, indicating the high stability of the complex formed by the PPO3–**4b** group.

The Gibbs FEL was computed using the Gromacs scripts g_sham and xpm2txt.py, and the RMSD and Rg values were utilized to calculate the Gibbs free energy. In the Gibbs FEL plots, the RMSD, Rg, and Gibbs free energy are depicted on the X, Y, and Z axes, illustrating favored conformations during dynamic simulations. Weak protein–ligand interactions are characterized by rough, multi-clustered FEL surfaces, while strong interactions exhibit smooth, single-clustered surfaces. Dark purple/blue spots in the FEL denote stable conformations, whereas red/yellow indicates instability. PPO3 complexes with both compound **4b** (Figure 8a) and the co-crystallized ligand (Figure 8b) exhibit single concentrated energy clusters, indicating stable interactions between PPO3 and compound **4b**.

Upon achieving stability in the complex systems, we computed the MM/PBSA binding free energies for the complexes of the PPO3 protein with compound **4b** (Figure 9a) and the co-crystallized ligand (Figure 9b). The average binding free energies of the PPO3 protein with compound **4b** and the co-crystallized ligand were −19.37 and −12.47 kcal/mol, respectively. This indicated that the PPO3 protein bound more strongly with compound **4b** than with the co-crystallized ligand.

In the residue energy analysis, compound **4b** showed optimal binding with the amino acid residues VAL-283 and HIS-263 in the PPO3 protein, with binding energies of -2.35 and −1.69 kcal/mol, respectively (Figure 10a). This highlights the significant roles of VAL-283 and HIS-263 in the interaction between compound **4b** and the PPO3 protein. Similarly, the co-crystallized ligand exhibited strong binding with the amino acid residues VAL-283, ASN-260, and HIS-263 in the PPO3 protein (Figure 10b), indicating their prominent involvement in the interaction. The residue energy results corroborate the findings from the docking simulations for compound **4b** and the co-crystallized ligand.

To explore the binding dynamics of the complex in pre- and post-molecular dynamics simulations, we aligned the structures of compound **4b** (Figure 11a) and the co-crystallized ligand (Figure 11b) with the PPO3 protein at 0, 25, 50, 75, and 100 ns from the simulation trajectories. The binding positions for both compounds showed minimal changes, underscoring the robust stability of these complexes.

## 3. Materials and Methods

### 3.1. Chemicals and Instruments

A melting-point apparatus (Shanghai YICE Equipment Co., Ltd., WRS-2, Shanghai, China) was used to determine the melting points (uncorrected) of the synthesized compounds. The synthesized compounds were analyzed using a Waters 2695 system (Waters Corp., Milford, MA, USA) equipped with a 250 × 4.6 mm Kromasil C18 column (Eka Nobel, Bohus, Sweden, EU). The ^1^H NMR spectra of the compounds were recorded on a JNM-ECX 400 (FT-NMR system, 400 MHz, JEOL Co., Akishima, Japan) with 400 MHz spectrophotometers using DMSO-*d6* as a solvent. HRMS was performed on a SYNAPT G2 mass spectrometer (Waters Corp., Manchester, UK) with positive ion mode, *m/z* 50–1200 range, and time-of-flight (TOF) analysis for accurate molecular weight determination. All commercially available chemicals and reagents were utilized without further purification. The reaction progress was monitored using thin-layer chromatography (TLC). Mushroom tyrosinase, L-tyrosine, and kojic acid were purchased from Sigma Chemicals (St. Louis, MO, USA). 

### 3.2. General Procedure for the Synthesis of the Compounds

#### 3.2.1. Procedure for the Synthesis of Indole-3-Carbaldehyde Derivatives (**2f**, **g**)

Indole-3-carbaldehyde (0.01 mol) was dissolved in dry acetone (20 mL). Potassium carbonate (0.025 mol) was then added to the solution, which was stirred at room temperature for 30 min. Subsequently, ethyl bromide or iodomethane (0.011 mol) in 10 mL of acetone was added dropwise to the reaction mixture while maintaining a temperature of 50 °C. The mixture was stirred for 3 h. Upon completion of the reaction, which was monitored using TLC, the reaction mixture was filtered to remove the formed salts. The filtrate was concentrated to yield a solid product, which was recrystallized from 95% ethanol to obtain the pure compound. In the intermediates **1a** and **2b**–**2k**, **2g** and **2f** are synthesized compounds, while the others are commercially available.

1-ethyl-1H-indole-3-carbaldehyde (**2f**)

Light-yellow solid; yield: 85%, m.p: 99–101°C. Appendix A: ^1^H NMR (400 MHz, DMSO-*d6*) *δ* 9.91 (s, 1H), 8.34 (s, 1H), 8.15–8.09 (m, 1H), 7.65–7.59 (m, 1H), 7.36–7.22 (m, 2H), 4.30 (q, *J* = 7.3 Hz, 2H), 1.42 (t, *J* = 7.2 Hz, 3H) [71].

1-methyl-1H-indole-3-carbaldehyde (**2g**)

Light-yellow solid; yield: 82%, m.p: 68–70 °C. Appendix A: ^1^H NMR (400 MHz, DMSO-*d6*) *δ* 9.90 (s, 1H), 8.27 (s, 1H), 8.10 (dt, *J* = 7.6, 1.1 Hz, 1H), 7.58 (dt, *J* = 8.2, 1.0 Hz, 1H), 7.33 (ddd, *J* = 8.2, 7.1, 1.4 Hz, 1H), 7.27 (ddd, *J* = 8.2, 7.1, 1.1 Hz, 1H), 3.89 (s, 3H) [72].

#### 3.2.2. Procedure for the Synthesis of Indole–Thiourea Derivatives (**4a**–**4k**)

The appropriate aldehyde (1.0 mmol) was dissolved in anhydrous ethanol (30 mL). Thiosemicarbazide (1.0 mmol) and 3 drops of acetic acid were added to the solution. The reaction mixture was refluxed for 10 h and monitored using TLC. Upon completion, the mixture was cooled to room temperature. The resulting precipitate was filtered, washed with ether, and recrystallized from ethanol to yield the corresponding indole–thiourea derivatives (**4a–4k**).

(E)-2-((1H-indol-3-yl)methylene)hydrazine-1-carbothioamide (**4a**)

Light-yellow solid; yield: 80%; mp: 232–234 °C. Appendix A: ^1^H NMR (400 MHz, DMSO-*d6*) *δ* 11.60 (s, 1H), 11.14 (s, 1H), 8.30 (s, 1H), 8.22 (dt, *J* = 7.8, 1.0 Hz, 1H), 8.09–7.97 (m, 1H), 7.81 (s, 1H), 7.46–7.35 (m, 2H), 7.19 (ddd, *J* = 8.2, 7.0, 1.3 Hz, 1H), 7.12 (ddd, *J* = 8.1, 7.0, 1.1 Hz, 1H). Appendix A: HR-ESI-MS (*m/z*): calculation (219.0704, (M+H)^+^); observed (219.0705, (M+H)^+^) [73].

(E)-2-((5-fluoro-1H-indol-3-yl)methylene)hydrazine-1-carbothioamide (**4b**)

Yellow solid; yield: 84%; mp: 207–209 °C. Appendix A: ^1^H NMR (400 MHz, DMSO-*d6*) *δ* 11.70 (s, 1H), 11.13 (s, 1H), 8.27 (s, 1H), 8.01 (d, *J* = 2.6 Hz, 1H), 7.98 (dd, *J* = 5.1, 2.3 Hz, 1H), 7.87 (s, 1H), 7.58 (s, 1H), 7.42 (dd, *J* = 8.8, 4.6 Hz, 1H), 7.03 (td, *J* = 9.2, 2.6 Hz, 1H). Appendix A: ^13^C NMR (101 MHz, DMSO-*d6*) *δ* 176.54, 158.0 (d, *J* = 234.3 Hz), 140.69, 133.70, 132.80, 124.15 (d, *J* = 11.1 Hz), 112.80 (d, *J* =10.0 Hz), 111.32 (d, *J* =4.0 Hz), 110.83 (d, *J* = 26.0 Hz), 107.12 (d, *J* = 24.0 Hz). Appendix A: HR-ESI-MS (*m/z*): calculation (237.0610, (M+H)^+^); observed (237.0611, (M+H)^+^).

(E)-2-((5-chloro-1H-indol-3-yl)methylene)hydrazine-1-carbothioamide (**4c**)

Yellow solid; yield: 80%; mp: 212–214 °C. Appendix A: ^1^H NMR (400 MHz, DMSO-*d6*) *δ* 11.81–11.76 (m, 1H), 11.17 (s, 1H), 8.27 (s, 1H), 8.20 (d, *J* = 2.1 Hz, 1H), 8.02 (s, 1H), 7.89 (d, *J* = 2.5 Hz, 1H), 7.56 (s, 1H), 7.44 (d, *J* = 8.6 Hz, 1H), 7.19 (dd, *J* = 8.6, 2.1 Hz, 1H). Appendix A: ^13^C NMR (101 MHz, DMSO-*d6*) *δ* 176.66, 140.50, 135.57, 132.48, 125.37, 124.91, 122.76, 121.00, 113.37, 110.90. Appendix A: HR-ESI-MS (*m/z*): calculation (253.0315, (M+H)^+^); observed (253.0314, (M+H)^+^).

(E)-2-((5-bromo-1H-indol-3-yl)methylene)hydrazine-1-carbothioamide (**4d**)

Yellow solid; yield: 88%; mp: 215–217 °C. Appendix A: ^1^H NMR (400 MHz, DMSO-*d6*) *δ* 11.82–11.77 (m, 1H), 11.17 (s, 1H), 8.31 (d, *J* = 2.0 Hz, 1H), 8.26 (s, 1H), 8.04–8.00 (m, 1H), 7.87 (d, *J* = 2.4 Hz, 1H), 7.54 (s, 1H), 7.40 (dd, *J* = 8.6, 0.6 Hz, 1H), 7.31 (dd, *J* = 8.6, 2.0 Hz, 1H). Appendix A: HR-ESI-MS (*m/z*): calculation (296.9810, (M+H)^+^); observed (296.9809, (M+H)^+^) [74].

(E)-2-((5-methoxy-1H-indol-3-yl)methylene)hydrazine-1-carbothioamide (**4e**)

Yellow solid; yield: 89%; mp: 230–232 °C. Appendix A: ^1^H NMR (400 MHz, DMSO-*d6*) *δ* 11.49 (d, *J* = 2.9 Hz, 1H), 11.18 (s, 1H), 8.28 (s, 1H), 7.99 (s, 1H), 7.77 (d, *J* = 2.9 Hz, 1H), 7.58 (d, *J* = 2.5 Hz, 1H), 7.37–7.29 (m, 2H), 6.84 (dd, *J* = 8.8, 2.5 Hz, 1H), 3.80 (s, 3H). Appendix A: ^13^C NMR (101 MHz, DMSO-*d6*) *δ* 176.52, 154.52, 141.09, 132.01, 131.35, 124.56, 112.53, 112.32, 110.80, 103.83, 55.31. Appendix A: HR-ESI-MS (*m/z*): calculation (249.0810, (M+H)^+^); observed (249.0809, (M+H)^+^).

(E)-2-((1-ethyl-1H-indol-3-yl)methylene)hydrazine-1-carbothioamide (**4f**)

Yellow solid; yield: 85%; mp: 188–190 °C. Appendix A: ^1^H NMR (400 MHz, DMSO-*d6*) *δ* 9.91 (s, 1H), 8.34 (s, 1H), 8.15–8.09 (m, 1H), 7.65–7.59 (m, 1H), 7.31 (ddd, *J* = 8.2, 7.0, 1.4 Hz, 1H), 7.26 (td, *J* = 7.4, 1.2 Hz, 1H), 4.30 (q, *J* = 7.3 Hz, 2H), 1.42 (t, *J* = 7.2 Hz, 3H). Appendix A: ^13^C NMR (101 MHz, DMSO-*d6*) *δ* 176.48, 140.43, 136.61, 133.08, 124.52, 122.69, 122.44, 120.87, 110.35, 110.15, 40.67, 15.19. Appendix A: HR-ESI-MS (*m/z*): calculation (247.1017, (M+H)^+^); observed (247.1018, (M+H)^+^).

(E)-2-((1-methyl-1H-indol-3-yl)methylene)hydrazine-1-carbothioamide (**4g**)

Yellow solid; yield: 85%; mp: 199–201°C. Appendix A: ^1^H NMR (400 MHz, DMSO-*d6*) *δ* 11.15 (s, 1H), 8.27 (s, 1H), 8.24 (dt, *J* = 7.9, 1.1 Hz, 1H), 8.05–8.00 (m, 1H), 7.80 (s, 1H), 7.48 (dt, *J* = 8.3, 1.0 Hz, 1H), 7.45–7.40 (m, 1H), 7.26 (ddd, *J* = 8.3, 7.1, 1.2 Hz, 1H), 7.16 (ddd, *J* = 8.0, 7.1, 1.1 Hz, 1H), 3.80 (s, 3H). Appendix A: HR-ESI-MS (*m/z*): calculation (233.0861, (M+H)^+^); observed (233.0861, (M+H)^+^) [75].

(E)-2-((5-methoxy-1-methyl-1H-indol-3-yl)methylene)hydrazine-1-carbothioamide (**4h**)

Yellow solid; yield: 80%; mp: 203–205 °C. Appendix A: ^1^H NMR (400 MHz, DMSO-*d6*) *δ* 11.15 (s, 1H), 8.25 (s, 1H), 7.98 (s, 1H), 7.75 (s, 1H), 7.59 (d, *J* = 2.5 Hz, 1H), 7.39 (d, *J* = 8.9 Hz, 1H), 7.36 (s, 1H), 6.90 (dd, *J* = 8.9, 2.5 Hz, 1H), 3.81 (s, 3H), 3.77 (s, 3H). Appendix A: ^13^C NMR (101 MHz, DMSO-*d6*) *δ* 176.51, 154.85, 140.68, 134.93, 132.76, 124.98, 112.29, 111.07, 109.66, 104.11, 55.41, 33.02. Appendix A: HR-ESI-MS (*m/z*): calculation (263.0967, (M+H)^+^); observed (263.0968, (M+H)^+^).

(E)-2-((5-bromo-1-methyl-1H-indol-3-yl)methylene)hydrazine-1-carbothioamide (**4i**)

Yellow solid; yield: 86%; mp: 220–222 °C. Appendix A: ^1^H NMR (400 MHz, DMSO-*d6*) *δ* 11.15 (s, 1H), 8.33 (d, *J* = 1.9 Hz, 1H), 8.23 (s, 1H), 8.03 (s, 1H), 7.86 (s, 1H), 7.60–7.55 (m, 1H), 7.48 (d, *J* = 8.7 Hz, 1H), 7.38 (dd, *J* = 8.7, 1.9 Hz, 1H), 3.80 (s, 3H). Appendix A: ^13^C NMR (101 MHz, DMSO-*d6*) *δ* 176.65, 139.97, 136.37, 135.84, 125.81, 125.35, 124.01, 113.81, 112.35, 109.74, 33.04. Appendix A: HR-ESI-MS (*m/z*): calculation (310.9966, (M+H)^+^); observed (310.9967, (M+H)^+^).

Methyl(E)-3-((2-carbamothioylhydrazineylidene)methyl)-1H-indole-5-carboxylate (**4j**)

Yellow solid; yield: 85%; mp: 247–249 °C. Appendix A: ^1^H NMR (400 MHz, DMSO-*d6*) *δ* 11.97 (s, 1H), 11.36 (s, 1H), 8.69 (s, 1H), 8.33 (s, 1H), 8.22 (s, 1H), 7.99 (d, *J* = 2.7 Hz, 1H), 7.81 (dd, *J* = 8.5, 1.6 Hz, 1H), 7.52 (d, *J* = 8.6 Hz, 1H), 7.21 (s, 1H), 3.85 (s, 3H). Appendix A: ^13^C NMR (101 MHz, DMSO-*d6*) *δ* 176.98, 167.08, 139.83, 139.56, 131.97, 123.85, 123.55, 123.43, 122.02, 112.15, 112.03, 51.86. Appendix A: HR-ESI-MS (*m/z*): calculation (277.0759, (M+H)^+^); observed (277.0759, (M+H)^+^).

(E)-2-((5-hydroxy-1H-indol-3-yl)methylene)hydrazine-1-carbothioamide (**4k**)

Light-gray solid, yield: 83%; mp: 261–263 °C. Appendix A: ^1^H NMR (400 MHz, DMSO-*d6*) *δ* 11.35 (d, *J* = 2.9 Hz, 1H), 11.22 (s, 1H), 8.88 (s, 1H), 8.23 (d, *J* = 5.4 Hz, 2H), 7.69 (d, *J* = 2.9 Hz, 1H), 7.50 (d, *J* = 2.4 Hz, 1H), 7.22 (d, *J* = 8.7 Hz, 1H), 7.07 (d, *J* = 3.4 Hz, 1H), 6.67 (dd, *J* = 8.7, 2.4 Hz, 1H). Appendix A: ^13^C NMR (101 MHz, DMSO-*d6*) *δ* 176.41, 152.13, 141.00, 131.22, 131.06, 124.88, 112.43, 112.29, 110.29, 105.87. Appendix A: HR-ESI-MS (*m/z*): calculation (235.0654, (M+H)^+^); observed (235.0657, (M+H)^+^).

### 3.3. Biology

#### 3.3.1. Tyrosinase Inhibition Assay

In the enzymatic assay, a reaction mixture was prepared containing 0.1 M potassium phosphate buffer (pH 6.8), 2 mM L-tyrosine, and 2500 units of tyrosinase. Samples at various concentrations were dispensed as 20 µL aliquots into a 96-well plate. Each well received 130 µL of substrate solution (buffer and L-tyrosine), followed by 5 µL of tyrosinase and 45 µL of buffer solution. The reaction mixture was then incubated at 37 °C for 10 min, after which absorbance at 490 nm was measured.

#### 3.3.2. Kinetic Mechanism Analysis

The kinetic experiment was intended to elucidate the inhibitory behavior of compound **4b** against tyrosinase. Four different doses of **4b** (0, 3, 6, and 12 µM) were investigated to characterize its inhibition pattern with respect to tyrosinase activity. Various concentrations of L-tyrosine substrate ranging from 0.1 to 1 mM were employed across all experiments. The initial incubation time and reading period mirrored those outlined in the anti-tyrosinase inhibition method. The maximum initial velocity (V₀) was determined from the linear phase of absorbance measurements taken over 5 min following the addition of the enzyme solution. The inhibition mode was further delineated using a Lineweaver–Burk plot, where the reciprocal of velocities (1/V) was plotted against the reciprocal of substrate concentrations utilized.

#### 3.3.3. Free Radical-Scavenging Assay

The DPPH radical-scavenging activity was evaluated using Blois’ method. A 0.2 mM solution of DPPH in ethanol (EtOH) was prepared as the working reagent. Each sample (20 μL) was introduced into a 96-well plate, followed by the addition of 180 μL of the DPPH solution. The reaction mixture was incubated at room temperature for 15 min. After the reaction, the absorbance was recorded at 515 nm using a microplate reader. The SC_50_ value, representing the concentration required to scavenge 50% of the DPPH radicals, was determined for each sample. Each measurement was performed in triplicate, and the mean value was reported. Ascorbic acid served as the positive control throughout the experiment.

### 3.4. Computational Methodology

#### 3.4.1. Molecular Properties and Drug Likeness

Pharmacokinetic parameters were evaluated by predicting the ADMET and drug-likeness properties using the compounds’ SMILES information. This analysis was conducted comprehensively using several models, including ADMETlab 3.0 [76], SwissADME (http://www.swissadme.ch/; accessed on 15 July 2024) [77], pkCSM (https://biosig.lab.uq.edu.au/pkcsm/; accessed on 15 July 2024) [78], and the admetSAR 2.0 web server [79].

#### 3.4.2. Molecular Docking Simulation

The receptor protein PPO3 mushroom tyrosinase (PDB ID: 2Y9X) and human tyrosinase-related protein 1 (PDB ID: 5M8M) were obtained from the PDB database, and their 3D structure files were downloaded. The protein structures were inspected using the PyMOL 3.0.3 software in preparation for molecular docking. All synthesized compounds were drawn in 3D using ACD/ChemSketch Freeware, and their structures were optimized using the MMFF94 force field in OpenBabel 2.0.2 software to obtain the most energetically favorable conformations. AutoDock Tools 1.5.6 was employed to prepare the proteins by adding hydrogen atoms and preprocess the ligand by hydrogenating and determining rotatable bonds. The grid parameters for molecular docking were set based on the crystallographic position of the ligand in the protein structures. The mTYR protein was set as follows: Center (X, Y, Z) = (−10.2, −30.3, −44.4), Size (X × Y × Z) = (15.0 × 15.0 × 15.0). The TYRP1 protein was set as follows: Center (X, Y, Z) = (−25.8, −26.1, 22.8), Size (X × Y × Z) = (15.0 × 15.0 × 15.0). Semi-flexible docking was performed with an exhaustiveness of 25 using the Lamarckian genetic algorithm in AutoDock Vina 1.2.0. The docking simulations yielded binding free energies and docking result files. Re-docking experiments were conducted using the co-crystallized ligand from the protein structures, and the root mean square deviation (RMSD) was calculated between the initial and re-docked poses. An RMSD value less than 2 Å indicated successful methodological validation. Additionally, to validate the reliability of molecular docking, the proteins were docked with the ligand under identical conditions in 10 independent runs, and the resulting data were compared to identify any differences.

#### 3.4.3. Molecular Dynamics (MD) Simulations

MD simulations of the PPO3 protein complexes with the co-crystallized ligand and compound **4b** were conducted using Gromacs 2022. The Charmm36 force field was applied for the protein, while Gaff2 was utilized for the ligand. The TIP3P water model was employed to solvate the protein–ligand systems within a periodic boundary box of 1.2 nm. To mimic realistic experimental conditions, sodium and chloride ions were added to neutralize the system’s charge. The simulation protocol consisted of three main stages of preparation. Initially, energy minimization was carried out using the steepest descent algorithm for 50,000 steps, terminating when the maximum force reached below 1000 kJ/mol. Subsequently, NVT pre-equilibration was performed for 50,000 steps at a constant temperature of 310 K, with a time step of 2 fs. This was followed by NPT pre-equilibration for 50,000 steps at a constant temperature of 310 K and pressure of 1 atm, also using a time step of 2 fs. After completing energy minimization and equilibration steps, a 100 ns MD simulation was conducted without constraints while employing a time step of 2 fs and saving coordinates every 10 ps.

## 4. Conclusions

In this study, a series of synthesized compounds (**4a**–**4k**) were evaluated for tyrosinase inhibition. Compound **4b** showed the highest inhibitory activity with an IC_50_ of 5.9 ± 2.47 μM, outperforming kojic acid (16.4 ± 3.53 μM). SAR analysis highlighted the importance of the fluorine substituent and unsubstituted amino group on the indole core. Compound **4k** demonstrated significant antioxidant activity due to its hydroxyl. Compound **4b** exhibited favorable ADMET and drug-likeness properties, positioning it as a more promising drug candidate compared to kojic acid and tropolone. 

Molecular docking of compounds **4a**–**4k** targeting tyrosinase revealed strong binding affinities for compounds **4a**, **4b**, **4f**, and **4k** to mTYR and TYRP1. Compound **4b** exhibited superior binding energies (−7.0 kcal/mol with mTYR and −6.5 kcal/mol with TYRP1) to those of tropolone and kojic acid, emphasizing its efficacy. In MD simulations, compound **4b** displayed stable interactions with PPO3 over 100 ns, which were characterized by low RMSD and RMSF values, indicating structural stability and minimal residue fluctuations. Rg analysis confirmed the consistent compactness of the PPO3–**4b** complex. The study demonstrated that compound **4b** forms a highly stable complex with PPO3, as evidenced by the consistent SASA curves, single-clustered Gibbs FEL surfaces, and stronger MM/PBSA binding free energy (−19.37 kcal/mol) than those of the co-crystallized ligand. Hydrogen bonding analysis showed persistent interactions with active site residues of tyrosinase, highlighting its reliable binding profile. Residue analysis confirmed robust interactions, particularly with VAL-283 and HIS-263, corroborating the docking simulation results. These findings provide valuable insights for the rational design and optimization of novel tyrosinase inhibitors with enhanced therapeutic potential. However, further in vivo studies are necessary to verify the efficacy and safety.

## Data Availability

Data are contained within the article and Appendix A.

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
