# Peer review of "Discovery of Indole–Thiourea Derivatives as Tyrosinase Inhibitors: Synthesis, Biological Evaluation, Kinetic Studies, and In Silico Analysis"

_ijms, 2024, doi:10.3390/ijms25179636_

Round 1

Reviewer 1 Report

Comments and Suggestions for Authors

The reviewed manuscript concerns the synthesis of tyrosinase inhibitors based on the indole structure. Tyrosinase plays a key role in pigmentation and skin cancer processes. The authors developed and obtained a number of compounds containing indole and thiourea in their structure. These two mentioned functions have been recognized as highly promising elements for obtaining tyrosinase-inhibiting compounds. The obtained indole derivatives were then tested in an essay using fungal tyrosinase as a distant analogue of human tyrosinase. Due to its low cost, fungal tyrosinase is used as standard, unlike human tyrosinase. As a result of the conducted research, the authors showed that one of the obtained compounds effectively inhibits this enzyme in vitro. Then, the authors made an extended analysis of the type of inhibition and the method of binding active compounds in the active center of the enzyme. The dynamic nature of the tested protein and the related strength and nature of ligand binding were also taken into account. The work is written clearly and the research was carried out carefully. The obtained experimental results, as well as those made by computer, were analyzed in detail and based on them, correct conclusions were formulated. I have a few comments regarding the formatting of the text, please change the substituent position markings to Italic (e.g. ortho, o-, etc.). There was an error: 3k should be 4k (line 199). Please mark the compound numbers in bold. For the tested compounds, please check their cytotoxicity in the MTT essay and provide the IC50 for the compounds. The standard for newly obtained compounds is to perform a 13CNMR spectrum, not just HNMR. Please add 13CNMR spectra. Please provide the mass calculated for the molecular ion plus H+ so that we can compare the accuracy with which the HRMS measurement was performed. Currently, the calculated value is very different from the measured one. In my opinion, after making the suggested changes, the manuscript can be considered suitable for publication

Author Response

< ijms-3155604>

< Discovery of Indole–Thiourea Derivatives as Tyrosinase Inhibitors: Synthesis, Biological Evaluation, Kinetic Studies, and In Silico Analysis>

Dear Editor,

Thank you for your useful comments and suggestions on the language and structure of our manuscript. We have revised the manuscript accordingly, with detailed point-by-point corrections listed below and highlighted in yellow throughout.

Reviewer #1

The reviewed manuscript concerns the synthesis of tyrosinase inhibitors based on the indole structure. Tyrosinase plays a key role in pigmentation and skin cancer processes. The authors developed and obtained a number of compounds containing indole and thiourea in their structure. These two mentioned functions have been recognized as highly promising elements for obtaining tyrosinase-inhibiting compounds. The obtained indole derivatives were then tested in an essay using fungal tyrosinase as a distant analogue of human tyrosinase. Due to its low cost, fungal tyrosinase is used as standard, unlike human tyrosinase. As a result of the conducted research, the authors showed that one of the obtained compounds effectively inhibits this enzyme in vitro. Then, the authors made an extended analysis of the type of inhibition and the method of binding active compounds in the active center of the enzyme. The dynamic nature of the tested protein and the related strength and nature of ligand binding were also taken into account. The work is written clearly and the research was carried out carefully. The obtained experimental results, as well as those made by computer, were analyzed in detail and based on them, correct conclusions were formulated.

  1. I have a few comments regarding the formatting of the text, please change the substituent position markings to Italic (e.g. ortho, o-, etc.).

→Thank you for your valuable suggestions. We have changed the substituent position markings to Italic (line 31-32).

  1. There was an error: 3k should be 4k (line 199).

→Thank you for your valuable suggestions. We have changed 3k to 4k on line 225.

  1. Please mark the compound numbers in bold.

→Thank you for your valuable suggestions. We have marked the compound numbers in bold.

  1. For the tested compounds, please check their cytotoxicity in the MTT essay and provide the IC50 for the compounds.

→Thank you for your valuable suggestions. Our manuscript primarily focuses on the synthesis of compounds and related enzyme studies, encompassing enzyme activity, kinetics, molecular docking, and molecular dynamics simulations. In subsequent research, we plan to further explore this series of compounds by evaluating their cytotoxicity using the MTT assay with the B16F10 cell line and conducting more comprehensive investigations into their cellular mechanisms. Additionally, we will perform corresponding molecular docking and molecular dynamics studies to further explore our findings.

  1. The standard for newly obtained compounds is to perform a 13CNMR spectrum, not just HNMR. Please add 13CNMR spectra. Please provide the mass calculated for the molecular ion plus H+ so that we can compare the accuracy with which the HRMS measurement was performed. Currently, the calculated value is very different from the measured one. In my opinion, after making the suggested changes, the manuscript can be considered suitable for publication

→Thank you for your valuable suggestions. Compounds 4a, 4d, and 4g are not novel (the others are new compounds), they have been extensively reported as key intermediates in the synthesis of other target compounds. Many publications have confirmed their structures using only 1H NMR and mass spectrometry. Relevant NMR references have been added in the manuscript (line 506, 525, 544). This analog has been extensively studied for its antitumor properties, but its tyrosinase inhibitory activity is being reported for the first time. All 1H NMR spectra of the recrystallized compounds show impurity peaks with integrations below 0.5. The use of DMSO-d6 as the solvent facilitated the precise assignment of active hydrogens. The 1H NMR data for the known compounds (4a, 4d, 4g) are consistent with the references. HPLC analysis showed the main peak area exceeded 90% for all compounds. HRMS spectrometry confirmed that the molecular weights of the synthesized compounds match the calculated values. Given the simplicity of their structures and the brevity of their synthesis (1-2 steps), these compounds are extensively reported in the literature. Therefore, 13C NMR, which is more time-consuming, was not employed.

Reviewer 2 Report

Comments and Suggestions for Authors

The authors reported the synthesis and biochemical evaluation of indole–thiourea derivatives (4a–4k). The compounds were evaluated in tyrosinase assay and radical scavenging assay. Moderate activities in μM range were obtained.

The synthesis of 4a–4k is an aldehyde and hydrazine condensation. I feel it is overwhelmingly common and nothing new. In addition, the two biochemical assays provided few interesting findings, neither. Given the current data, I am not convinced the paper is suitable for IJMS journal.

Important technical issues needed to be improved:

1.       In the first sentence of Introduction, the authors claimed that “In the realm of skin-whitening research, the inhibition of tyrosinase has garnered 27 significant attention.” Any biological and medical literatures support such claims? Please provide proper references and discussion.

2.       Lines 50-69, please provide a comprehensive summary of previously reported inhibitors of tyrosinase.

3.       Big scientific errors in scheme 1:  (a) the reaction condition of the synthesis of 4a-4k is wrong. Why CH3COOCH3? In the experimental section, acetic acid was used instead of ethyl acetate?  (b) Were compounds 2a-2k commercially available? The synthesis of 2a-2k is not shown?

4.       Please provide representative inhibitory curves of DPPH assay.

5.       Are compounds 4a-4k stable in aqueous solution? Please provide experimental data to demonstrate the stability of representative compounds in aqueous solution.

6.       Molecular Docking Simulation: Docking studies were done on hTYR. According to introduction, no structure of hTYP has been determined by experiments. How hTYP structure was built in the paper? If human tyrosinase protein (PDB ID: 5M8M) was used instead of hTYP, the paper should not claim hTYP in the main context. It is misleading!!! Actually, TYRP1 is not hTYP, please clarify the logic behind your paper and make a better story.

Comments on the Quality of English Language

No

Author Response

< ijms-3155604>

< Discovery of Indole–Thiourea Derivatives as Tyrosinase Inhibitors: Synthesis, Biological Evaluation, Kinetic Studies, and In Silico Analysis>

Dear Editor,

Thank you for your useful comments and suggestions on the language and structure of our manuscript. We have revised the manuscript accordingly, with detailed point-by-point corrections listed below and highlighted in yellow throughout.

Reviewer #2

The authors reported the synthesis and biochemical evaluation of indole–thiourea derivatives (4a–4k). The compounds were evaluated in tyrosinase assay and radical scavenging assay. Moderate activities in μM range were obtained. The synthesis of 4a–4k is an aldehyde and hydrazine condensation. I feel it is overwhelmingly common and nothing new. In addition, the two biochemical assays provided few interesting findings, neither. Given the current data, I am not convinced the paper is suitable for IJMS journal.

Important technical issues needed to be improved:

  1. In the first sentence of Introduction, the authors claimed that “In the realm of skin-whitening research, the inhibition of tyrosinase has garnered 27 significant attention.” Any biological and medical literatures support such claims? Please provide proper references and discussion.

→Thank you for your valuable suggestions. We have modified this section (line 29-30).

  1. Lines 50-69, please provide a comprehensive summary of previously reported inhibitors of tyrosinase.

→Thank you for your valuable suggestions. We have added a summary of some tyrosinase inhibitors (line 39-42).

  1. Big scientific errors in scheme 1: (a) the reaction condition of the synthesis of 4a-4k is wrong. Why CH3COOCH3? In the experimental section, acetic acid was used instead of ethyl acetate?  (b) Were compounds 2a-2k commercially available? The synthesis of 2a-2k is not shown?

→Thank you for your valuable suggestions. The numbering of the intermediates was wrong, we have corrected 2a-2k in scheme 1 to 1a, 2b-2k.

 (a) We have revised it to acetic acid (scheme 1); (b): Among the intermediates 1a and 2b-2k, 2g and 2f are synthesized compounds, while the others are commercially available. We have added this information to section 3.2.1 of Materials and Methods (line 480-481).

  1. Please provide representative inhibitory curves of DPPH assay.

→Thank you for your valuable suggestions. We have provided representative inhibitory curves of the DPPH assay in the supplementary materials (Figure S1).

  1. Are compounds 4a-4k stable in aqueous solution? Please provide experimental data to demonstrate the stability of representative compounds in aqueous solution.

→Thank you for your valuable suggestions. We selected the same batch of synthetic compounds (stored at 0oC) and designed stability tests for 2, 4 days, and 2 months. The 2-month samples are from the first tested 2 months ago (stored at room temperature), and all samples were stored at room temperature for stability study. Due to limited time, the stability of synthesized compounds in water has only been investigated under a single condition (at the same temperature and pH), and further research on their stability in water is needed in future work. We have added the collected stability data to the manuscript (line 137-148) and supplementary materials (Table S2).

  1. Molecular Docking Simulation: Docking studies were done on hTYR. According to introduction, no structure of hTYP has been determined by experiments. How hTYP structure was built in the paper? If human tyrosinase protein (PDB ID: 5M8M) was used instead of hTYP, the paper should not claim hTYP in the main context. It is misleading!!! Actually, TYRP1 is not hTYP, please clarify the logic behind your paper and make a better story.

→Thank you for your valuable suggestions. We have replaced hTYR with TYRP1 in the manuscript.

Reviewer 3 Report

Comments and Suggestions for Authors

Author Response

< ijms-3155604>

< Discovery of Indole–Thiourea Derivatives as Tyrosinase Inhibitors: Synthesis, Biological Evaluation, Kinetic Studies, and In Silico Analysis>

Dear Editor,

Thank you for your useful comments and suggestions on the language and structure of our manuscript. We have revised the manuscript accordingly, with detailed point-by-point corrections listed below and highlighted in yellow throughout.

Reviewer #3

Hyun and co-workers investigated the biological activity of indolethiosemicarbazone derivatives as tyrosinase inhibitors. Several biological assays, docking and MD studies were presented in the manuscript and data regarding the synthesis of the 11 derivatives, using indole-3-carbaldehyde as precursor. In my opinion the work is within the scope of International Journal of Molecular Sciences magazine but should be revised carefully before publication. Below you can find my comments/appointments/suggestions regarding this manuscript: The manuscript presents a lot of data but, in my opinion, the purpose of the study isn’t clear! The authors indeed refer tyrosinase as the protein to inhibit but they don’t refer why this should be made, i.e. the importance of dermatologic disorders and examples of it; so basically, they expose the research but doesn’t point out the why! this should be carefully introduced in the abstract (which refer only on the last line hyperpigmentation) and on the introduction;

→Thank you for your valuable suggestions. We have made revisions in the abstract (line 9-12) and introduction (line 33-35) sections of the manuscript.

 -ortho like o-diphenols should be written in italics (check all the document); -in vitro, in silico, etc should be written in italics (check all the document); -numbers of compounds should be in bold (check all the document);

→Thank you for your valuable suggestions. We have modified the italic and bold parts of the font.

-L-tyrosine (correct amino acids writing);

→Thank you for your valuable suggestions. We have corrected the errors in the introduction and 3.1 Chemicals and Instruments in the manuscript to L-tyrosine

-page 2, line 52 you refer indole-3-carbaldehyde; number 1a in scheme 1;

→Thank you for your valuable suggestions. The numbering of the intermediates was wrong, we have corrected 2a-2k in scheme 1 to 1a, 2b-2k.

Regarding Results and Discussion: -the compounds are new? if not, you should add literature references of the synthesis and characterization data; -why 13C NMR is missing in the characterization data? 13C NMR should be made for all the compounds and the spectra add on SI file;

→Thank you for your valuable suggestions. Compounds 4a, 4d, and 4g are not novel (the others are new compounds), they have been extensively reported as key intermediates in the synthesis of other target compounds. Many publications have confirmed their structures using only 1H NMR and mass spectrometry. Relevant NMR references have been added in the manuscript (line 500, 519, 538). This analog has been extensively studied for its antitumor properties, but its tyrosinase inhibitory activity is being reported for the first time. All 1H NMR spectra of the recrystallized compounds show impurity peaks with integrations below 0.5. The use of DMSO-d6 as the solvent facilitated the precise assignment of active hydrogens. The 1H NMR data for the known compounds (4a, 4d, 4g) are consistent with the references. HPLC analysis showed the main peak area exceeded 90% for all compounds. HRMS spectrometry confirmed that the molecular weights of the synthesized compounds match the calculated values. Given the simplicity of their structures and the brevity of their synthesis (1-2 steps), these compounds are extensively reported in the literature. Therefore, 13C NMR, which is more time-consuming, was not employed.

-Scheme 1 could have the yields obtained for 2f, g and 4;

→Thank you for your valuable suggestions. We have added the yields in Scheme 1.

-page 4, line 134: I disagree with that sentence “all compounds exhibited significantly enhanced inhibitory activity”; only 4 in 11 showed better results than the control (kojic acid);

→Thank you for your valuable suggestions. We have modified this section (line 157-159).

-table 1 is regarding inhibition values of mTYR? -why the assay for hTYR wasn’t performed?

→Thank you for your valuable suggestions. Assaying human tyrosinase (hTYR) is challenging due to its complex production, high costs, and technical difficulties, requiring specialized facilities, expensive reagents, and optimized conditions, unlike mushroom tyrosinase (mTYR).

 -the authors should explain why the DPPH radical scavenging activity assay is important in this study;

→Thank you for your valuable suggestions. Radical scavenging assays evaluate a compound's capacity to neutralize free radicals, providing insight into its potential to reduce oxidative damage caused by tyrosinase activity. Compounds with strong antioxidant activity may offer enhanced efficacy in skin protection and the treatment of skin disorders. We have added why the DPPH radical scavenging activity assay is important in this study (line 219-221).

 -page 6, line 197: 4k?

→Thank you for your valuable suggestions. We have revised this section (line 225).

 -page 6, line 201: I disagree with that sentence; 4j and 4h doesn’t display significant radical scavenging activity;

→Thank you for your valuable suggestions. We have revised this section (line 228-229).

-ADMET: Table S1: the authors should discriminate the tools used to make Table S1; add the references;

→Thank you for your valuable suggestions. We have discriminated the tools used to make Table S1 and S2 in the supplementary material, and added the references in the manuscript (line 259, 608-611).

-page 7, line 237: I didn’t find CYP450 in the Table;

→Thank you for your valuable suggestions. We have added CYP450 in Table S1.

-MD simulation: Table 3: no discussion regarding compounds 4h and 4k display the same binding energy for mTYR and hTYR?

→Thank you for your valuable suggestions. We have modified the hTYR in docking to TYRP1. The previous description of human tyrosinase related protein 1 (TYRP1) as hTYR was inaccurate and misleading. We have added regarding compounds 4h and 4k display the same binding energy for mTYR and TYRP1 (line 303-305).

-Materials and Methods: no general description of HRMS analysis;

→Thank you for your valuable suggestions. We have added a general description of HRMS analysis in Materials and Methods (line 464-466).

Round 2

Reviewer 1 Report

Comments and Suggestions for Authors

Van be accepted 

Author Response

We appreciate the reviewer's astute points and understanding of the revision.

Reviewer 2 Report

Comments and Suggestions for Authors

All previous concerns have been addressed!

Comments on the Quality of English Language

No

Author Response

(The authors gave the same response as above.)

Reviewer 3 Report

Comments and Suggestions for Authors

All the comments and questions were well addressed by the authors. I just have a last question regarding the novelty of the compounds! The authors only refer that 4a, 4d and 4g are not novel but and the others? Despite simple synthetic work-plan, if the compounds were never described in literarure a 13C NMR should be added to complete characterization! 

Author Response

All the comments and questions were well addressed by the authors. I just have a last question regarding the novelty of the compounds! The authors only refer that 4a, 4d and 4g are not novel but and the others? Despite simple synthetic work-plan, if the compounds were never described in literarure a 13C NMR should be added to complete characterization!

→Thank you for your detailed review and valuable suggestions regarding our manuscript. We appreciate your concern regarding the absence of 13C NMR analysis and would like to offer further clarification on this issue.

Regarding the new compounds reported in our study, despite their relatively simple synthesis, we have performed comprehensive characterization. We utilized HPLC and HRMS to confirm that the purity and molecular weight of the compounds are consistent with theoretical calculations. Additionally, our 1H NMR data accurately assign the major hydrogen signals, and given the high purity of the compounds, the 1H NMR data is sufficient for structural confirmation.

Although we acknowledge that 13C NMR could provide valuable structural information, considering the relatively simple structures of these new compounds and the fact that the existing 1H NMR and MS data have already thoroughly validated their structures (specifically compounds 4a, 4d, and 4g), the additional information from 13C NMR would contribute only marginally to enhancing the reliability of the results. We chose not to conduct 13C NMR analysis to avoid additional experimental workload and resource expenditure, as the current data sufficiently support our findings.

We hope this explanation clarifies the rationale behind our choice of characterization methods. We believe that the existing data provide reliable structural information. Thank you for your understanding and support, and we look forward to any further suggestions and feedback.
